# MIRACLE3D: Memory-efficient Integrated Robust Approach for Continual Learning on Point Clouds via Shape Model Construction

**Hossein Resani & Behrooz Nasihatkon**
K. N. Toosi University of Technology
`hossein.resani@gmail.com`
`nasihatkon@kntu.ac.ir`

## Abstract

In this paper, we introduce a novel framework for memory-efficient and privacy-preserving continual learning in 3D object classification. Unlike conventional memory-based approaches in continual learning that require storing numerous exemplars, our method constructs a compact shape model for each class, retaining only the mean shape along with a few key modes of variation. This strategy not only enables the generation of diverse training samples while drastically reducing memory usage but also enhances privacy by eliminating the need to store original data. To further improve model robustness against input variations—an issue common in 3D domains due to the absence of strong backbones and limited training data—we incorporate *Gradient Mode Regularization*. This technique enhances model stability and broadens classification margins, resulting in accuracy improvements. We validate our approach through extensive experiments on the ModelNet40, ShapeNet, and ScanNet datasets, where we achieve state-of-the-art performance. Notably, our method consumes only 15% of the memory required by competing methods on the ModelNet40 and ShapeNet, while achieving comparable performance on the challenging ScanNet dataset with just 8.5% of the memory. These results underscore the scalability, effectiveness, and privacy-preserving strengths of our framework for 3D object classification.

## 1 Introduction

Continual learning is an approach in machine learning where models are designed to learn and adapt incrementally from new data over time without forgetting previously acquired knowledge. One of the central challenges in this context is *catastrophic forgetting*—the tendency of neural networks to lose previously learned information when adapting to new tasks or data. While significant progress has been made in addressing catastrophic forgetting in 2D images, its application to 3D point clouds is still relatively underdeveloped and presents distinct challenges. Unlike the structured grids of 2D images, 3D point clouds consist of irregularly spaced points, requiring specialized techniques such as PointNet (Qi et al., 2017) and graph neural networks (Wang et al., 2019) for effective feature extraction. Moreover, the 3D domain lacks large-scale datasets similar to ImageNet (Deng et al., 2009), limiting the ability of models to learn robust features. These challenges are exacerbated in replay-based continual learning methods, where traditional exemplar selection techniques, such as Herding (Rebuffi et al., 2017; Welling, 2009), struggle with the irregular and multimodal nature of 3D feature spaces. This makes it particularly challenging to retain previously learned features while acquiring new information, heightening the risk of catastrophic forgetting. Addressing these issues in continual learning for 3D point clouds is crucial for enabling adaptive, resilient systems in real-world applications such as autonomous driving, robotics, and augmented reality, where models must continuously learn from an evolving stream of sensory data while retaining their ability to accurately recognize past patterns and structures.

In continual learning, a common strategy involves replay-based methods (Rebuffi et al., 2017; Wu et al., 2019; Belouadah & Popescu, 2018; Chen et al., 2020), where small subsets of data from each task, called *exemplars*, are stored and later replayed alongside current data during training on

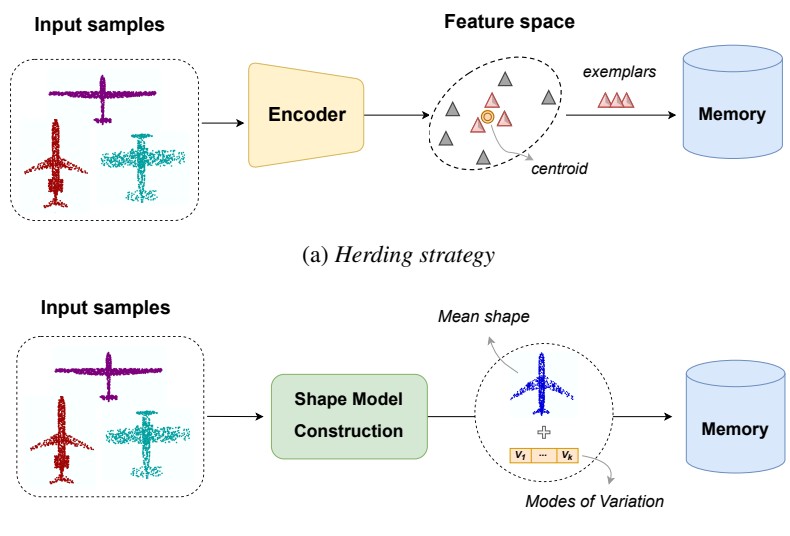

(a) *Herding strategy*

(b) *Our method*

Figure 1: Comparison of the overview of (a) previous methods and (b) our method. Previous methods depend on a herding strategy (Rebuffi et al., 2017; Welling, 2009), which is less effective for 3D point clouds.

new tasks. These exemplars are important because they help mitigate catastrophic forgetting by allowing the model to maintain a connection to past knowledge while adapting to new information. However, traditional approaches in 2D and 3D continual learning that require storing raw input data from previous tasks introduce significant challenges, especially in the context of privacy and memory efficiency. Storing raw data can raise legal concerns under regulations like the European GDPR, where users have the right to request the deletion of their personal data, making compliance difficult if original data must be retained for learning purposes. This issue becomes even more critical when working with sensitive information, such as medical imaging, where the potential for privacy violations is high. Moreover, storing exemplars over multiple tasks leads to an increasing memory burden, which is impractical, especially for real-time applications where computational resources are limited. While some 2D continual learning approaches attempt to address this by storing low-dimensional features instead of full images (Iscen et al., 2020) or using low-fidelity images (Zhao et al., 2021), this issue remains relatively underexplored in 3D continual learning.

To address these challenges, we propose a novel exemplar generation method focusing on memory efficiency, privacy preservation, and backbone independence. Our approach utilizes the geometric properties of input point clouds to create a compact shape model, storing only the mean shape along with a few modes of variation. This not only preserves privacy but also significantly reduces memory requirements. Additionally, we introduce Gradient Mode Regularization to enhance robustness against input variations within the shape space. Our approach is also backbone-independent, allowing easy integration with different architectures. Our contributions can be summarized as:

- **Memory-Efficient and Privacy-Preserving Shape Generation:** Using a shape model to represent an entire class of objects leads to a highly compact representation. As a side effect, it avoids the storage of original data.

- **Gradient Mode Regularization:** Penalizing the gradient of the loss along the principal modes of shape variation enhances the representational capabilities of samples drawn from shape models.

- **State-of-the-Art Performance:** Achieving state-of-the-art performance on three major point cloud benchmarks: ModelNet40 (Wu et al., 2015), ShapeNet (Chang et al., 2015), and ScanNet (Dai et al., 2017). In the final incremental session, it achieves improvements of 4.9%, 2.1%, and 1.8% on ModelNet40, ShapeNet, and ScanNet, respectively, while using only 15% of the memory compared to the state-of-the-art for ModelNet40 and ShapeNet, and just 8.5% of the memory for ScanNet.

## 2 RELATED WORK

### 2.1 SHAPE REPRESENTATION VIA STATISTICAL SHAPE MODELS

Shape representation involves methods that describe the geometry and structure of an object, enabling efficient analysis, manipulation, and comparison. Statistical Shape Models (SSMs) are a widely recognized approach for shape representation, providing a low-dimensional, parametric model of complex objects. Typically, SSMs are constructed by applying Principal Component Analysis (PCA) to a set of objects in dense correspondence. This yields a linear model in which shapes are represented as points in a low-dimensional affine vector space. While non-linear and multi-linear variants exist, PCA-based SSMs remain the most commonly used in practice due to their simplicity and effectiveness. (Cootes et al., 1995). SSMs are extensively used in computer vision, particularly for modeling human-related features such as faces, bodies (Ambellan et al., 2019), bones (Rajamani et al., 2004; Sarkalkan et al., 2014), and organs (Weiherer et al., 2023). Their applications span across numerous fields, including face recognition (Blanz & Vetter, 2003; Feng et al., 2021; Li et al., 2023; Zielonka et al., 2022) and body reconstruction, as well as various areas in medicine (Fouefack et al., 2020; Lüthi et al., 2017; Weiherer et al., 2023), forensics, cognitive science, neuroscience, and psychology (Egger et al., 2020a). Recent advancements (Loiseau et al., 2021; Raju et al., 2022), explore new methodologies for improving shape representations, including geometric deep learning techniques and improved handling of non-linearities. In this study, we develop a compact shape model for each class by preserving only the mean shape and a select number of key variation modes.

### 2.2 3D CONTINUAL LEARNING

Despite significant progress in 2D continual learning, 3D continual learning remains relatively underdeveloped. Several recent work aim to address challenges such as catastrophic forgetting and data sparsity. Chowdhury et al. (2021) introduced a method leveraging knowledge distillation and semantic word vectors to mitigate the forgetting of prior training. Zhao & Lee (2022) addressed class-incremental 3D object detection by using a static teacher for pseudo annotations of old classes and a dynamic teacher to continually learn new data. Zamorski et al. (2023) proposed Random Compression Rehearsal (RCR), which employs a compact, autoencoder-like model to compress and store critical data from previous tasks. Camuffo & Milani (2023) developed a continual learning approach for semantic segmentation in LiDAR point clouds, tackling coarse-to-fine segmentation challenges and data sparsity issues. Chowdhury et al. (2022) proposed using *Microshapes*—orthogonal basis vectors to describe 3D objects—to help align features between synthetic and real data, reducing domain gaps and enhancing robustness against real-world noise. I3DOL (Dong et al., 2021) uses an adaptive-geometric approach to handle irregular point clouds and an attention mechanism to focus on significant geometric structures, aiming to minimize forgetting. It also introduces a fairness compensation strategy to balance training between new and old classes. Later, InOR-Net (Dong et al., 2023) improved on this by enhancing geometric feature extraction and incorporating a dual fairness strategy to effectively manage class imbalances and prevent biased predictions. However, both methods still suffer from inefficiencies, as they rely on the herding approach (Rebuffi et al., 2017; Welling, 2009), which struggles to effectively capture diversity in 3D point clouds—a crucial issue highlighted by Resani et al. (2024). Addressing this limitation by developing an exemplar selection approach specifically designed to handle the unique characteristics of 3D data remains an important and underexplored research area.

CL3D (Resani et al., 2024) demonstrates that the herding approach (Rebuffi et al., 2017; Welling, 2009), commonly used in the 2D domain, is unsuitable for the 3D domain. It introduces a novel exemplar selection strategy for 3D continual learning, utilizing spectral clustering and combining input, local, and global features, achieving state-of-the-art performance. While the use of raw input features offers independence from network-specific dependencies, the CL3D method still relies on network-derived features for local and global clustering. Moreover, storing original data in memory raises concerns regarding privacy preservation, which serves as a major motivation for our work. In our approach, we propose a novel *exemplar generation* strategy that avoids storing original data, thereby ensuring privacy preservation.

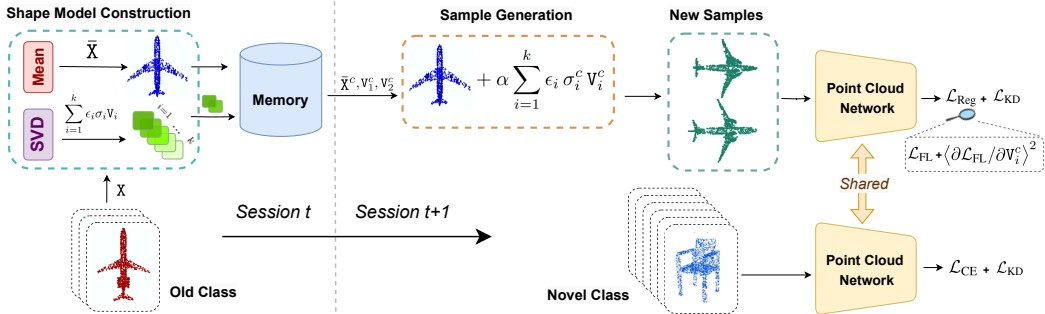

Figure 2: **Overview of our method.** For each old class, a compact shape model is created using the mean shape and a few key modes of variation derived from SVD, which are stored in memory. In the next session, new samples are generated by applying perturbations to the mean shape using these modes. These generated samples, along with novel class data, are processed through a shared point cloud network, trained using cross-entropy, knowledge distillation, and Gradient Mode Regularization to enhance robustness and prevent forgetting.

## 3 REPLAY-BASED CONTINUAL LEARNING SETTING

Consider a sequence of disjoint tasks $\mathcal{D} = \{\mathcal{D}^1, ..., \mathcal{D}^T\}$, and let $\mathcal{C}^t = \{c_1^t, ..., c_{m_t}^t\}$ be the set of target classes in task $\mathcal{D}^t$. We assume that the classes between all tasks are disjoint, i.e., $\mathcal{C}^i \cap \mathcal{C}^j = \emptyset$ for $i \neq j$. The goal of continual learning is to progressively train a model, where at each session $t$, only the training samples from the current task $\mathcal{D}^t = \{(X^t, Y^t)\}$, consisting of point clouds $X^t$ and their corresponding labels $Y^t$, are accessible. During testing, the model trained on task $\mathcal{D}^t$ is expected to predict outputs not only for the current task but also for all prior tasks $\mathcal{D}^1, ..., \mathcal{D}^{t-1}$.

### 3.1 EXEMPLAR MEMORY MANAGEMENT

In continual learning, memory management for storing exemplars typically follows two strategies (Zhou et al., 2023). The first is to maintain a fixed number of samples per class, ensuring consistent representation as new tasks are added, though leading to a linearly increasing memory size. The second is to impose a fixed overall memory cap, which stabilizes memory usage but may reduce class representation as more tasks accumulate. Here, we adopt the first strategy.

## 4 PROPOSED METHOD

Our approach develops a memory-efficient, privacy-preserving continual learning model for 3D point cloud classification. By storing only the mean shape and key variations for each class, we significantly reduce memory usage. Gradient Mode Regularization further enhances robustness during incremental sessions. An overview of *MIRACLE3D* is shown in fig. 2, and we provide further details in the following sections.

### 4.1 PRELIMINARIES

In morphable shape models (Egger et al., 2020b), samples of an object class are represented as a *mean shape* plus a linear combination of several basis shape modes. Consider a set of point clouds $X_1, X_2, \ldots, X_m$, each being a sample from a specific object class. These point clouds are assumed to be pre-registered and resampled, so that each contains exactly $n$ points, with all points in correspondence. Hence, each point cloud sample can be expressed by an $n \times 3$ matrix $X_i \in \mathbb{R}^{n \times 3}$. The shape model comprises the mean shape $\bar{X} = \frac{1}{m} \sum_{i=1}^{m} X_i \in \mathbb{R}^{n \times 3}$, principal modes of variation $V_1, V_2, \ldots, V_k \in \mathbb{R}^{n \times 3}$, and their corresponding scales $\sigma_1, \sigma_2, \ldots, \sigma_k \in \mathbb{R}$. To derive these modes, we vectorize the mean-centered samples, resulting in $\mathbf{y}_i = \text{vect}(X_i - \bar{X}) \in \mathbb{R}^{3n}$. The modes $V_i$ are then obtained as the first $k$ left singular vectors of the matrix $Y = [\mathbf{y}_1, \mathbf{y}_2, \ldots, \mathbf{y}_m] \in \mathbb{R}^{3n \times m}$, which are subsequently reshaped into $n \times 3$ matrices. The scalars $\sigma_i$ are the associated singular values. In many cases, the samples $X_i$ are assumed to be normally distributed. Thus, a new shape can be

generated as:

$$\mathtt{X} = \bar{\mathtt{X}} + \sum_{i=1}^{k} \epsilon_i \sigma_i \mathtt{V}_i, \tag{1}$$

where $\epsilon_i \in \mathbb{R}$ are sampled from a standard normal distribution.

## 4.2 PROTOTYPES AS SHAPE MODELS

The main idea of our approach is to represent the samples of an object class $c$ using a compact shape model. This shape model is defined by three main components: the mean shape $\bar{\mathtt{X}}^c$, a set of modes of variation $\mathtt{V}_1^c, \mathtt{V}_2^c, \dots, \mathtt{V}_k^c \in \mathbb{R}^{n \times 3}$, and the associated scaling factors $\sigma_1^c, \sigma_2^c, \dots, \sigma_k^c \in \mathbb{R}$, where the superscript $c$ indicates that the model belongs to class $c$. Here, $k$ is kept small to maintain memory efficiency while capturing the essential variability within the class. The modes of variation, together with the scaling factors, capture the key dimensions along which the shape of class samples can vary, allowing for a compact representation of the shape diversity. When novel classes are introduced in the continual learning scenario, we need to generate samples from previously learned classes to mitigate catastrophic forgetting. To accomplish this, we generate synthetic samples from the old class $c$ using

$$\mathtt{Z}^c = \bar{\mathtt{X}}^c + \alpha \sum_{i=1}^{k} \epsilon_i \, \sigma_i^c \, \mathtt{V}_i^c. \tag{2}$$

This equation is similar to the more general form presented in (1) but includes an adjustment: a scalar coefficient $\alpha < 1$. This coefficient is deliberately chosen to be small (around 0.2) in order to restrict the generated samples to a high-probability region of the shape space. By using a smaller value of $\alpha$, we ensure that the generated shapes represent the most typical and probable configurations of the class. For each old class $c$, we generate $n_s$ synthetic samples $\{\mathtt{Z}_j^c\}_{j=1}^{n_s}$. These generated samples serve as a proxy for the original data of the old classes, and they are used alongside the novel class samples to retrain the model, thereby preventing it from forgetting previously acquired knowledge. Note that the samples are not stored in memory but are generated on demand. This process allows the model to maintain its performance across both old and novel classes, ensuring a more balanced and memory-efficient continual learning capability. The results of shape modeling, along with examples of generated point cloud samples, are presented in fig. 4.

## 4.3 LOSS FUNCTION

Our loss function is defined as

$$\mathcal{L}(\theta) = \sum_{c \in \mathcal{C}_{\text{new}}} \sum_{\substack{j \\ \mathbf{y}_j = c}} \left( \mathcal{L}_{\text{CE}}(\theta, \mathtt{X}_j, c) + \mathcal{L}_{\text{KD}}(\theta, \mathtt{X}_j, c) \right) + \sum_{c \in \mathcal{C}_{\text{old}}} \sum_{j=1}^{n_s} \mathcal{L}_{\text{FL}}(\theta, \mathtt{Z}_j^c, c), \tag{3}$$

where, $\theta$ is the network parameters, $\mathcal{C}_{\text{new}}$ and $\mathcal{C}_{\text{old}}$ are the sets of new and old classes, respectively, $\mathcal{L}_{\text{CE}}$ is the cross-entropy loss, $\mathcal{L}_{\text{KD}}$ is the knowledge distillation loss (Li & Hoiem, 2017), and $\mathcal{L}_{\text{FL}}$ is the focal loss (Lin et al., 2017)

$$\mathcal{L}_{\text{FL}}(\theta, \mathtt{Z}_j^c, c) = -\alpha_t \left( 1 - P_c(\theta, \mathtt{Z}_j) \right)^\gamma \log(P_c(\theta, \mathtt{Z}_j)), \tag{4}$$

where $P_c(\theta, \mathtt{Z}_i)$ is the output softmax probability of the network for class $c$. As suggested in Resani et al. (2024), we apply focal loss to tackle the class imbalance issue, given that the number of samples $n_s$ for the old classes is generally significantly smaller than the number of samples for the new classes.

## 4.4 GRADIENT MODE REGULARIZATION

Regularizing the gradient norm of a neural network's output with respect to its inputs is the idea of *Double Backpropagation* (Drucker & Le Cun, 1991), aimed at reducing sensitivity and promoting stability in model learning (Varga et al., 2017). This technique helps control output changes in response to input variations, thus enhancing robustness against adversarial attacks (Lee et al., 2022;

Ross & Doshi-Velez, 2018; Finlay & Oberman, 2021) and improving generalization (Rame et al., 2022), particularly with small training datasets. Our idea here is to perform gradient regularization in the directions of the major modes of variation $\mathbb{V}_i^c$.

Consider a sample $\mathbb{Z}_j^c$ derived from a shape model using equation (2). When a small perturbation is introduced along the direction of each mode of variation $\mathbb{V}_i^c$, it is crucial to ensure that the resulting shape maintains a low loss value. To ensure this, for the old class samples $\mathbb{Z}_j^c$, we replace the focal loss $\mathcal{L}_{\text{FL}}(\theta, \mathbb{Z}_j^c, c)$ in (3) with a regularized loss:

$$\mathcal{L}_{\text{Reg}}(\theta, \mathbb{Z}_j^c, c) = \mathcal{L}_{\text{FL}}(\theta, \mathbb{Z}_j^c, c) + \lambda \sum_{i=1}^{k} \left\langle \partial \mathcal{L}_{\text{FL}}(\theta, \mathbb{Z}_j^c, c)/\partial \mathbb{Z}_j^c \, , \, \mathbb{V}_i^c \right\rangle^2 , \quad (5)$$

where $\langle \cdot, \cdot \rangle$ denotes the inner product, $\partial \mathcal{L}_{\text{FL}}(\theta, \mathbb{Z}_j^c, c)/\partial \mathbb{Z}_j^c$ represents the gradient of the loss function with respect to the network input $\mathbb{Z}_j^c$, and $\lambda$ is a balancing hyperparameter. The added regularization term encourages the directional derivative of the cost function to be close to zero in the direction of the modes $\mathbb{V}_i^c$, which helps ensure that the modified shape remains consistent with the original shape distribution after small perturbations along these modes. fig. 3 illustrates our Gradient Mode Regularization concept. In the absence of Gradient Mode Regularization, variations in input and small perturbations can lead to decreased efficiency, pushing the model away from the low-error region of previously learned classes. By applying Gradient Mode Regularization, our approach preserves a larger margin from the low-error region of old classes, thereby enhancing robustness and stability against such changes.

## 5 EXPERIMENTS

We adopt the experimental setup used by I3DOL (Dong et al., 2021), InOR-net (Dong et al., 2023), and CL3D (Resani et al., 2024), including the datasets, backbone architecture, and the number of incremental sessions. To the best of our knowledge, these are the only existing studies that focus on exemplar-based continual learning for point clouds, making them the most suitable for comparison with our results.

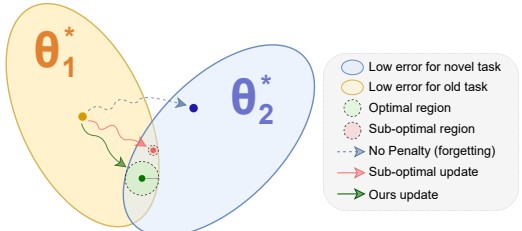

Figure 3: **Illustration of Gradient Mode Regularization.** The figure illustrates the parameter spaces $\theta_1^*$ and $\theta_2^*$, highlighting the low-error regions for the old and novel classes, respectively. Without any regularization, model updates (blue arrow) may lead to catastrophic forgetting. Additionally, the absence of regularization can result in sub-optimal updates (red arrow), making the model sensitive to small perturbations, even within the shape space. In contrast, our method (green arrow) employs Gradient Mode Regularization to steer updates toward a more reliable region for the old classes, providing a larger margin than previous methods. This approach improves the model's stability and robustness against perturbations during continual learning.

**Datasets.** We conduct our evaluation using three datasets: ModelNet40 (Wu et al., 2015), ShapeNet (Chang et al., 2015), and ScanNet (Dai et al., 2017). ModelNet40 contains 40 classes of clean 3D CAD models. In alignment with Dong et al. (2021) and Dong et al. (2023), we utilize 10 incremental sessions, each adding four new classes. Our method stores the mean shape of each class alongside two modes of variation, resulting in three components stored per class. This results in a total memory usage of 120 samples—representing an 85% reduction compared to methods I3DOL and InOR-Net—highlighting the memory efficiency aspect of our approach.

For the ShapeNet dataset, we use 53 categories, consistent with I3DOL and InOR-net Dong et al. (2023). We follow their configuration of nine incremental sessions, with each session introducing six new classes, except for the final session which introduces five classes. Since the ShapeNet dataset is already aligned, no alignment preprocessing is required, and only resampling is performed.

Table 1: Accuracy comparison on the ModelNet40 (Wu et al., 2015), ShapeNet (Chang et al., 2015), and ScanNet (Dai et al., 2017) datasets. The table reports the average accuracy across all sessions (Avg.), the accuracy of the last session (Last), and the total number of exemplars stored in memory ($M$). Notably, unlike other approaches, our method does not store any original samples. The term "*Joint*" represents learning all classes simultaneously (upper bound), while "*FT*" indicates sequential learning of sessions without any forgetting mitigation (lower bound). Best results are highlighted in **bold** and second-best results are underlined. The table is an extended version of Dong et al. (2023)

| | ModelNet40 | | | ShapeNet | | | ScanNet | | |
|---|---|---|---|---|---|---|---|---|---|
| **Method** | $A_{avg}$ | $A_{last}$ | $M$ | $A_{avg}$ | $A_{last}$ | $M$ | $A_{avg}$ | $A_{last}$ | $M$ |
| *Joint (Upper bound)* | 94.3 | 88.5 | *all* | 93.3 | 89.3 | *all* | 93.0 | 91.0 | *all* |
| *FT (Lower bound)* | 28.0 | 9.2 | 0 | 26.9 | 6.2 | 0 | 30.1 | 8.2 | 0 |
| *2D Approaches Applied on 3D Point Clouds* | | | | | | | | | |
| LwF (Li & Hoiem, 2017) | 60.3 | 31.5 | - | 63.4 | 39.5 | - | 53.1 | 38.1 | - |
| iCaRL (Rebuffi et al., 2017) | 68.9 | 39.6 | 800 | 69.5 | 44.6 | 1000 | 56.0 | 36.3 | 600 |
| DeeSIL (Belouadah & Popescu, 2018) | 72.1 | 43.7 | 800 | 71.7 | 47.2 | 1000 | 63.1 | 43.7 | 600 |
| EEIL (Castro et al., 2018) | 75.0 | 48.1 | 800 | 74.2 | 51.6 | 1000 | 65.1 | 45.7 | 600 |
| IL2M (Belouadah & Popescu, 2019) | 81.1 | 57.6 | 800 | 77.6 | 61.4 | 1000 | 66.7 | 48.3 | 600 |
| DGMw (Chen et al., 2020) | 77.2 | 53.4 | 800 | 73.8 | 49.2 | 1000 | 63.6 | 43.8 | 600 |
| DGMa (Chen et al., 2020) | 76.8 | 51.5 | 800 | 73.4 | 48.7 | 1000 | 63.7 | 44.7 | 600 |
| BiC (Wu et al., 2019) | 48.5 | 66.8 | 800 | 78.8 | 64.2 | 1000 | 66.8 | 48.5 | 600 |
| RPS-Net (Rajasegaran et al., 2019) | 81.7 | 58.3 | 800 | 78.4 | 63.5 | 1000 | 67.3 | 49.1 | 600 |
| *3D Specific Methods* | | | | | | | | | |
| I3DOL (Dong et al., 2021) | 85.3 | 61.5 | 800 | 81.6 | 67.3 | 1000 | 70.2 | 52.1 | 600 |
| InOR-Net (Dong et al., 2023) | 87.0 | 63.9 | 800 | 83.7 | 69.4 | 1000 | 72.2 | 54.8 | 600 |
| CL3D (Resani et al., 2024) | 85.1 | 69.7 | 120 | 80.8 | 67.6 | 120 | 71.8 | 55.4 | 120 |
| **MIRACLE3D (Ours)** | **88.1** | **74.6** | **120** | **84.9** | **71.5** | **120** | **72.9** | **57.2** | **120** |

The ScanNet dataset consists of 17 classes collected from real indoor scene scans. We use the FilterReg (Gao & Tedrake, 2019) to register the samples before resampling them. Our approach stores three components per class. We conduct nine incremental sessions, each introducing two new classes, with the final session adding just one class.

**Evaluation Metric and baselines.** In table 1, We report the average accuracy after the final session, denoted as $A_{last}$, along with the average incremental accuracy, denoted as $A_{avg}$, which represents the average of accuracies after all sessions (including the initial one). For a more complete analysis, we include the scenario where the network has access to the entire dataset from previous tasks (joint training), which serves as an ideal upper bound. Conversely, as a lower bound, we provide results for the scenario involving complete forgetting (denoted as FT in the tables), where the model updates its parameters exclusively based on new tasks. Furthermore, we include the results of well-known 2D approaches applied to 3D point clouds to demonstrate their shortcomings in this context, emphasizing the necessity for dedicated 3D-specific methods.

**Implementation Details.** *MIRACLE3D* is implemented in PyTorch and trained on a Tesla V4 GPU with a batch size of 32. We use PointNet Qi et al. (2017) as the backbone for fair comparison with prior work (Dong et al., 2021; 2023), though our approach is not tied to any specific architecture. The Adam optimizer is used with 50 epochs per incremental session. Focal loss addresses data imbalance between old and novel classes, and the distillation factor is set to 0.1. We generate 10 samples per class during training, which are not stored in memory.

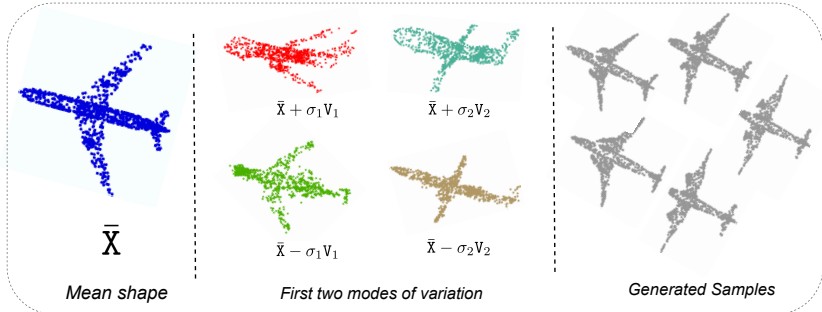

Figure 4: **Results of Shape Modeling:** Mean shape (left), first two principal modes of variation (center), and examples of generated point cloud samples (right).

## 5.1 COMPARISON

**Results.** table 1 demonstrates a comparison on 3 datasets including ModelNet40 dataset (Wu et al., 2015), ShapeNet dataset (Chang et al., 2015) and ScanNet (Dai et al., 2017). In the ModelNet40 dataset, our proposed method, *MIRACLE3D*, outperforms all competing approaches in terms of both average accuracy ($A_{avg}$) and the accuracy of the last session ($A_{last}$), achieving 88.1% and 74.6%, respectively. These results indicate a significant improvement over other 3D-specific methods like I3DOL and InOR-Net, which achieved lower average accuracies of 85.3% and 87.1%, respectively. Our memory usage (M) is notably efficient, requiring only 120 exemplars, compared to the 800 or 1000 exemplars used by other methods.

The ShapeNet dataset (Chang et al., 2015) includes more classes compared to ModelNet40 and ScanNet, making it a more complex scenario. On this dataset, *MIRACLE3D* achieves an average accuracy of 84.9% and a last session accuracy of 71.5%, outperforming other 3D-specific methods.

The ScanNet dataset (Dai et al., 2017), being a real-world point cloud dataset, introduces further challenges due to its inherent noisiness, incompleteness, and other complex characteristics typical of real-world scans. Despite these challenges, *MIRACLE3D* achieved an average accuracy of 72.9% and a last session accuracy of 57.2%, outperforming all other approaches evaluated, including CL3D and InOR-Net, which achieved average accuracies of 70.0% and 72.2%, respectively. Moreover, the adapted 2D approaches, such as RPS-Net, fell behind significantly, indicating that methods designed specifically for 3D point clouds are necessary for effective learning in real-world scenarios. Importantly, *MIRACLE3D* remains memory efficient with only 120 exemplars compared to the larger memory footprints of other methods.

## 5.2 ABLATIONS AND ANALYSIS

### 5.2.1 EFFECT OF GRADIENT MODE REGULARIZATION

The effect of Gradient Mode Regularization is evaluated to determine its impact on model performance in our setting. As shown in table 2, incorporating this regularization led to improvements in both average accuracy ($A_{avg}$) and last session accuracy ($A_{last}$). Specifically, Gradient Mode Regularization helped stabilize the model by reducing its sensitivity to input variations along key modes of shape variation, thereby enhancing robustness against the inherent variability of 3D point clouds.

Table 2: Comparison of average accuracy ($A_{avg}$) and last session accuracy ($A_{last}$) with and without Gradient Mode Regularization on the ModelNet40, ShapeNet and ScanNet datatsets.

| Method | ModelNet40 | | ShapeNet | | ScanNet | |
|---|---|---|---|---|---|---|
| | $A_{avg}$ | $A_{last}$ | $A_{avg}$ | $A_{last}$ | $A_{avg}$ | $A_{last}$ |
| *W/o Gradient Mode Regularization* | 87.5 | 72.3 | 84.1 | 69.4 | 72.3 | 56.1 |
| *W Gradient Mode Regularization* | **88.1** | **74.6** | **84.9** | **71.5** | **72.9** | **57.2** |

### 5.2.2 Optimal Number of Variation Modes

A core focus of our approach is optimizing memory efficiency, aiming to achieve minimal memory usage while outperforming state-of-the-art methods. Through extensive experimentation with different numbers of modes of variation, as shown in table 3, we found that configurations with 2 or 4 modes consistently offered superior performance compared to other configurations. However, given our emphasis on minimizing memory consumption, we chose to store only 2 modes of variation along with the mean shape. It's important to note that the optimal number of modes can vary depending on the variability within object classes and the quality of the shape models.

Table 3: Comparison of average accuracy ($A_{avg}$) and last session accuracy ($A_{last}$) using different combinations of shape components on the ModelNet40 dataset.

| Mean Shape | 1st | 2nd | 3rd | 4th | $A_{avg}$ (%) | $A_{last}$ (%) |
|:---:|:---:|:---:|:---:|:---:|:---:|:---:|
| ✓ | | | | | 81.3 | 66.8 |
| ✓ | ✓ | | | | 84.5 | 69.5 |
| ✓ | ✓ | ✓ | | | **88.1** | 74.6 |
| ✓ | ✓ | ✓ | ✓ | | 87.6 | 72.1 |
| ✓ | ✓ | ✓ | ✓ | ✓ | **88.1** | **74.8** |

### 5.2.3 Number of Generated Samples

We conducted experiments to determine the optimal number of generated samples as shown in table 4. We found that 10 samples yielded the best performance, surpassing other configurations. It is important to note that these samples are not stored in memory; rather, they are generated on-the-fly solely for training purposes. When we increased the number of samples from 10 to 20, we did not observe any improvement in performance. The lack of improvement may be attributed to an increase in the number of degraded samples within certain classes (as will be briefly discussed in the discussion), which could negatively impact the learning process.

Table 4: Comparison of average accuracy ($A_{avg}$) and last session accuracy ($A_{last}$) with varying numbers of generated samples on the ModelNet40 dataset.

| No. of generated samples | $A_{avg}$ (%) | $A_{last}$ (%) |
|:---|:---:|:---:|
| 5 samples | 85.9 | 69.8 |
| 10 samples | **88.1** | **74.6** |
| 20 samples | 87.3 | 72.3 |

### 5.2.4 Performance Across other Backbones

We adopted PointNet (Qi et al., 2017) as the backbone for our model to ensure a fair comparison with existing work in 3D continual learning (Dong et al. (2021; 2023); Resani et al. (2024)). A key advantage of our approach, however, lies in its backbone independence. Since our model operates directly in the input space, it can be seamlessly integrated with other backbones. To demonstrate this, we conducted additional experiments using more recent backbones, specifically DGCNN (Wang et al. (2019)) and PointMLP-elite (Ma et al. (2022)), as shown in table 5. One of the reported metrics, the Forgetting Rate (FR), measures the performance gap between joint training (the upper bound) and the model's actual performance. Our results show that because our method does not rely on backbone-specific features, the forgetting rate remains relatively consistent across different architectures. These findings highlight that our method achieves similar effectiveness regardless of the backbone, reinforcing the backbone independence of our approach.

Table 5: Comparison of performance across different backbones in terms of average accuracy (Ours $A_{avg}$), last session accuracy (Ours $A_{last}$), and forgetting rate (FR). Joint training results are included as the upper bound. The relatively consistent FR across different backbones highlights the backbone independence of our model.

| Backbones | Ours ($A_{avg}$) | Ours ($A_{last}$) | Joint ($A_{avg}$) | Joint ($A_{last}$) | FR (%)↓ |
|---|---|---|---|---|---|
| PointNet (Qi et al., 2017) | 88.1 | 74.6 | 94.3 | 88.5 | **6.2** |
| DGCNN (Wang et al., 2019) | 89.1 | 79.8 | 95.4 | 92.1 | **6.3** |
| PointMLP-elite (Ma et al., 2022) | 89.3 | 82.3 | 95.8 | 93.0 | **6.5** |

### 5.2.5 RANDOMNESS EFFECT IN SHAPE MODEL

The scalar coefficient $\alpha$ introduced in eq. (2) acts as a control variable that adjusts the level of variation in the generated samples. We deliberately set $\alpha$ to a small value to ensure that the generated samples remain within a high-probability region of the shape space. By using a smaller value for $\alpha$, the generated shapes are more likely to represent the most typical and probable configurations for each class. Through our experimentation in table 6, we found that an optimal value for this hyperparameter in our setting is $\alpha = 0.2$, which effectively balances the trade-off between diversity and representativeness.

Table 6: Comparison of average accuracy ($A_{avg}$) and last session accuracy ($A_{last}$) with three different levels of randomness: low (0.1, 0.2), medium (0.5), and high (1).

| Randomness weight | $A_{avg}$ (%) | $A_{last}$ (%) |
|---|---|---|
| 0.1 | 87.9 | 73.2 |
| **0.2** | **88.1** | **74.6** |
| 0.5 | 85.8 | 68.9 |
| 1 | 83.6 | 65.4 |

## 6 DISCUSSION

Our *MIRACLE3D* model demonstrates impressive capabilities; however, certain challenges persist, particularly due to the variability within some classes, which complicates smooth morphing between dissimilar samples. Integrating clustering methods could help by grouping more similar samples together, as suggested by Resani et al. (2024). Additionally, since our approach is based on point clouds, ensuring registration accuracy is crucial. Our current affine methods occasionally produce nonsensical and inaccurate shapes, although even these contribute to mitigating forgetting. This indicates that adopting nonlinear deformable registration could yield better results. Another promising direction is developing a direct neural network-based shape model (Loiseau et al., 2021), which presents challenges in terms of memory efficiency. To address these challenges, techniques like knowledge distillation (Hinton et al., 2015) could help reduce storage requirements for neural representations, making this approach more viable.

## 7 CONCLUSION

In this paper, we presented *MIRACLE3D*, a novel memory-efficient, privacy-preserving continual learning approach tailored for 3D point cloud classification. By constructing a compact shape model that captures the mean and key modes of variation for each class, our method significantly reduces memory requirements while avoiding the need to store raw data. Additionally, we introduced *Gradient Mode Regularization* to enhance model robustness, effectively mitigating catastrophic forgetting even under challenging input variations. A key strength of our method lies in its backbone independence, as it operates directly in the input space, enabling seamless integration with various 3D deep learning architectures. Our experiments, conducted across the ModelNet40, ShapeNet, and ScanNet datasets, demonstrated that *MIRACLE3D* achieves state-of-the-art performance, outperforming existing 3D-specific continual learning methods with just a fraction of the memory, regardless of the backbone used.

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
