# OpenReview forum: "MIRACLE 3D: Memory-efficient Integrated Robust Approach for Continual Learning on 3D Point Clouds via Shape Model Construction"
_ICLR.cc/2025/Conference — ICLR 2025 Poster_

### Official Review · Reviewer_9uF5 · 2024-11-01

**Soundness:** 3
**Presentation:** 3
**Contribution:** 3
**Rating:** 6
**Confidence:** 4

**Summary:**

This paper introduces a novel continual learning approach for 3D object classification. The key idea is to retain only the mean shape and a low number of modes for each of the previous classes, so that in the following tasks in the continual learning pipeline, samples from the previous classes can be conveniently generated instead of being saved in memory for replay.
This allows reducing the required memory at training time but also enhances privacy since there is no need to store the original old 3D shapes.
Furthermore, the authors propose a new regularization technique to steer the network parameters toward a better optimal solution and avoid catastrophic forgetting. The authors asses their contributions on three benchmarks based on the popular ModelNet40, ShapeNet, and ScanNet datasets, achieving state-of-the-art performance.

**Strengths:**

1. I consider the proposed method easy and, at the same time, highly effective. In my view, this is the main strength of the proposed approach.
2. The proposed approach shows great performance improvement over previous works.

**Weaknesses:**

1)The main weakness I can think of in this paper is on the presentation/writing side.
First of all, the listed contributions on page 2 are not really contributions, at least not all of them. For example, Geometry-Awareness and Backbone Independence are more properties than real contributions. I believe the authors should spend a consistent effort in rewriting this section and clearly state their two contributions, which are the memory-efficient and privacy-preserving shape generation and the Gradient Mode Regularization.
Similar considerations can be done about the related works. In my opinion, in some sections such as Sec. 2.1, 2.2, 2.4, it is not clear where the paper stands compared to previous works. Please make an effort to emphasize why these papers are related and why they differ.

2)A second problem, but still related to the previous one, is the lack of visualizations that can greatly improve the understanding of the paper. For example, why not show the generated samples while varying some parameters such as alpha and sigma? I would like to see these shapes to also have a qualitative idea of the proposed approach.

3)Lack of some details/ablations. How are the samples for each class selected? Can the authors provide some ablations regarding this aspect? Also, the contribution of the Gradient Mode Regularization seems to be marginal. It is okay as the paper is not entirely focused on this contribution; however, it would be important to show that it also works in other datasets and not only for ModelNet.

**Questions:**

I encourage the authors to work on the listed limitations, as they can me simply solved with some additional experiments/ablations and some writing.

---

### Official Review · Reviewer_d8L5 · 2024-11-03

**Soundness:** 3
**Presentation:** 2
**Contribution:** 3
**Rating:** 6
**Confidence:** 3

**Summary:**

The paper also draws on ideas from 2D continual learning approaches, particularly techniques that leverage low-fidelity images to enable memory-efficient learning. By borrowing these concepts, MIRACLE 3D adapts low-dimensional representation strategies to 3D point cloud data, tackling the unique challenges posed by 3D continual learning. It enables continual learning for 3D object classification with a focus on memory efficiency and privacy preservation.

**Strengths:**

The approach does not require the storage of original point cloud data, thus enhancing privacy.
By storing only a shape model (mean shape and modes of variation), the method achieves significant memory savings, reportedly using only 15% and 8.5% of the memory compared to competing methods on ModelNet40 and ScanNet, respectively.
The framework uses statistical shape models to capture the intrinsic geometric properties of 3D point clouds, which is particularly effective given the unstructured nature of 3D data.

**Weaknesses:**

The method’s reliance on storing mean shapes and a few modes of variation per class assumes limited intra-class variation, which may not hold in real-world applications with highly diverse categories. As the number of classes grows, even a few modes per class could lead to increased memory demands and possibly a drop in classification performance if the shape diversity is too restricted.
The paper lacks a deep exploration of how key hyperparameters, such as the number of modes of variation, RBF kernel parameters, or the coefficient in Gradient Mode Regularization, affect performance. Given the complexity of continual learning in 3D, tuning these parameters is likely essential to the model’s performance.

**Questions:**

[1] Could the authors display the mean shape along with a few primary modes of variation for selected classes? This visualization could clarify the model’s representational capacity and robustness for each class and would make it easier to evaluate how well the shape model captures intra-class diversity.

[2] Can the authors evaluate MIRACLE 3D on datasets with greater intra-class variation, such as SemanticKITTI or nuScenes, to assess scalability? This would provide a stronger validation of MIRACLE 3D’s scalability and potential adaptations for applications with high variability, giving a more comprehensive view of its robustness in real-world settings.

[3] Would the authors conduct a more thorough ablation study focusing on key hyperparameters, specifically the number of modes and gradient regularization strength?

[4] Could the authors explore additional regularization or data augmentation techniques to improve robustness against noise and occlusions?

---

### Official Review · Reviewer_DwN5 · 2024-11-04

**Soundness:** 2
**Presentation:** 2
**Contribution:** 2
**Rating:** 6
**Confidence:** 3

**Summary:**

The authors propose an exemplar generation method for memory-efficient and privacy-preserving continual learning in 3D object classification. Additionally, Gradient Mode Regularization is introduced to enhance robustness against input variations.

**Strengths:**

- The proposed method achieves state-of-the-art results on widely-used datasets.
- Compared with previous methods, MIRACLE3D is memory-efficient.

**Weaknesses:**

- Lack of ablation study. Only the Gradient Mode Regularization is evaluated on ModelNet.
- How does Geometry-awareness benefit for this task?
- The authors claim MIRACLE3D is backbone independent. However, all the experiments are conducted only on pointnet.

**Questions:**

see weakness.

---

> ### Comment · Reviewer_DwN5 · 2024-11-26
>
> Thanks for the response. Most of my concerns have been addressed. I will raise my score.

---

### Official Review · Reviewer_HZdC · 2024-11-04

**Soundness:** 3
**Presentation:** 3
**Contribution:** 2
**Rating:** 6
**Confidence:** 3

**Summary:**

MIRACLE 3D proposes to learn shape models from point cloud data for more efficient representation of data in continual learning scenarios. The authors use the shape model mean + variations to learn the data for a continual setting and then perform gradient regularization to promote stability of gradients to keep a balance for old and new classes. The compact representation leads to significant reduction in memory footprint. The method is evaluated on 3 popular datasets- ModelNet40, ScanNet, and ShapeNet where it performs well on classification.

**Strengths:**

* The paper is mostly well written and the method is clearly explained
* Significantly reduced memory footprint compared to exisiting methods as it uses shape models to represent the data
* An ablation is performed and the paper also explores optimal number of modes and number of samples to generate for continual learning

**Weaknesses:**

* The paper talks at length about continual larning in a setting which has multiple tasks but the proposed method is only shown for classification where new classes are learned. This makes the pitch of the paper seem confusing with the results. Moreover, the case for learning new classes is much weaker compared to learning new tasks (+ classes). Can the authors explore learning new tasks such as part segmentation andnobject retrieval?
* The case for using a shape model to reduce memory footprint for continual learning as the main contribution is weak in my opinion
* The proposed gradient mode regularization leads to minimal improvement in accuracy for classification. It could be that the extra stability is particularly helpful for learning new tasks in which case such a scenario should be tested. Metrics other than accuracy could help to make a more copelling case.

Post rebuttal:
See reply.

**Questions:**

* Learning a new task + exploring if the order of tasks matter in point cloud data might be interesting to see

---

> ### Comment · Reviewer_HZdC · 2024-11-23
>
> Thank you for the clarifications.
>
> The issue with the term "task" was indeed a misunderstanding. I would urge the authors to revise this part to clearly define what constitutes a new task. The order of learning tasks is interesting and could be a worthy addition in the supplemental.
> I'm mostly satisfied with the response and also read the text changes and additional experiments requested by other reviews, therefore, I'm happy to increase my score.

---

### Meta-Review · Area_Chair_RYh1 · 2024-12-23

**Metareview:**

The paper presents a method for continual learning for 3D object classification that is privacy-preserving and memory-efficient. While discussions on continual learning and efficiency are well-motivated, privacy preservation is not thoroughly justified through theory or experiments. It is recommended to reduce emphasis on privacy in the final version of the paper. But the paper overall presents an effective method and is clearly presented. The AC agrees with the reviewer consensus to recommend the paper for acceptance at ICLR.

**Additional Comments On Reviewer Discussion:**

DwN5 requires additional experiments for ablation and backbone choices, which are provided by the response leading to a score raised towards acceptance. HZdC acknowledges the clarifications on terminology presented in the author response. Results on additional datasets required by 9uF5 are presented in the author response. Authors acknowledge the generalization issue raised by d8L5 and provide additional experiments on hyperparameters, which are deemed sufficient by the reviewer.

---

### Decision · Program_Chairs · 2025-01-22

Accept (Poster)